# Application of Pulsed Electric Fields to Pilot and Industrial Scale Virgin Olive Oil Extraction: Impact on Organoleptic and Functional Quality

**DOI:** 10.3390/foods11142022

**Published:** 2022-07-08

**Authors:** Alberto Navarro, María-Victoria Ruiz-Méndez, Carlos Sanz, Melchor Martínez, Duarte Rego, Ana G. Pérez

**Affiliations:** 1Instituto de la Grasa (CSIC), Campus Universidad Pablo de Olavide, Edificio 46, Ctra. de Utrera, km 1, 41013 Seville, Spain; agprduran@gmail.com (A.N.); mvruiz@ig.csic.es (M.-V.R.-M.); carlos.sanz@ig.csic.es (C.S.); 2Acesur, Carretera de La Carolina, km 29, 23220 Vilches, Spain; mmartinez@acesur.com; 3EnergyPulse Systems, Est Paco Lumiar Polo Tecnológico Lt3, 1600-546 Lisbon, Portugal; duarte.rego@energypulsesystems.com

**Keywords:** olive oil processing, innovative technology, pulsed electric field, olive oil quality, phenolic compounds

## Abstract

The quality of virgin olive oil (VOO) is largely determined by the technology used in the industrial process of extracting the oil. Technological innovations within this field aim to strike a proper balance between oil yield and the optimal chemical composition of VOO. The application of pulsed electric fields (PEF) that cause the electroporation of the plant cell membranes favors a more efficient breakage of the olive fruit tissue, which in turn could facilitate the extraction of the oil and some of its key minor components. Pilot-scale and industrial extraction tests have been conducted to assess the effect of PEF technology on the oil extraction yield and on the organoleptic and functional quality of VOO. The best results were obtained by combining the PEF treatment (2 kV/cm) with short malaxation times and a low processing temperature. Under these conditions, PEF technology could decisively improve the oil yield by up to 25% under optimal conditions and enhance the incorporation of phenolic and volatile compounds into the oils. The PEF treatment neither affected the physicochemical parameters used to determine the commercial categories of olive oils, nor the tocopherol content. Similarly, the sensory evaluation of the PEF-extracted oils by means of a panel test did not detect the appearance of any defect or off-flavor. In addition, the intensity of positive attributes (fruity, bitter and pungent) was generally higher in PEF oils than in control oils.

## 1. Introduction

Virgin olive oil (VOO) is defined as the oil obtained from the fruit of the olive tree (*Olea europaea* L.) exclusively by mechanical or other physical means under conditions, particularly thermal conditions, that do not lead to alterations in the oil [1]. VOO is mainly composed of triglycerides rich in monounsaturated fatty acids; it is also an important source of nutritionally valuable compounds such as tocopherols (Vit E), phenolic compounds and pigments (chlorophylls and carotenoids) that are directly extracted from the olive fruit or formed during the oil extraction process from precursors initially present in the olive fruit. Some of these minor compounds, representing less than 2% of the oil, are related to the health-promoting properties of VOO—in particular with regard to its protective effects against the states of chronic inflammation and derived diseases [2]—and also determine VOO organoleptic quality [3]. Among them, it is important to highlight the role of VOO phenolic compounds, including both phenolic compounds with secoiridoid structures and tocopherols (vitamin E), that exhibit potent antioxidant properties related to the aforementioned benefits for human health [4,5]. Thus, health claims have been approved by the European Food Safety Authority (EFSA) regarding the evidence of the protective effect of VOO phenolics for cardiovascular diseases [6]—which may be applied to oils containing at least 250 ppm of hydroxytyrosol and derivatives—and the relationship between the dietary intake of vitamin E and protection from oxidative damage [7].

The extraction process of VOO, which includes milling of the fruits, olive paste malaxation and oil separation by means of pressing or centrifugation [1], largely determines the content of these minor compounds in the oil. Due to their effect on the chemical composition of the oil, milling and malaxation are the critical steps in the process [8]. During milling, the fruit cellular structures are broken, and the oil stored in the vacuoles of the mesocarp cells is released. In addition, due to this loss of cellular integrity, enzymes and substrates that were previously separated in different cellular compartments come into contact. A number of biochemical reactions are triggered, some of which are critical for the formation of key compounds for the functional and organoleptic properties of VOO, such as phenolic and volatile compounds [9,10]. Malaxation is necessary to break down the emulsion created by milling and, through coalescence phenomena, it helps to unite the small, dispersed oil droplets, forming larger ones that help increase the extraction yield. During malaxation, some of the biochemical processes initiated in the milling step remain active and the new compounds formed are distributed between the aqueous and oil phases. However, the dominant reactions in this stage seem to be chemical oxidation reactions favored by the continuous incorporation of oxygen into the olive paste. Thus, increasing malaxation time usually renders higher extraction yields but it may also have a negative impact on the oil’s organoleptic quality [11]. For this reason, research is being conducted to increase the efficiency of the extraction process with mild malaxation conditions or no malaxation at all. 

New technologies are being evaluated to achieve a more efficient breaking of the fruit tissue, facilitating the extraction of the oil without causing emulsions, and at the same time achieving a better extraction of key compounds for the functional and organoleptic quality of the oil. Among them is pulsed electric fields (PEF), which is based on the application of high-intensity electric fields in the form of pulses of short duration [12]. These pulses increase the intercellular potential of the cells causing the formation of pores on the cell membrane and/or a permanent disruption of the cell depending on the electric field intensity. Although PEF technology is defined as a non-thermal process, it may cause punctual and transitory increases in temperature that, however, are much lower than those induced by other technologies and do not seem to affect the nutritional properties of the treated foods [13]. The use of PEF technology for the improvement of the quality of different food products and beverages has been linked to its potential to improve organoleptic and functional properties such as color, texture, flavor, or antioxidants and vitamin contents [12]. Thus, PEF pretreatments have been successfully used to enhance the extractability of numerous bioactive compounds from different plant materials [14]. 

In the last few years, some promising results have been published on the benefits of using PEF to increase the oil yield in the industrial process to extract VOO. Some authors have reported oil yield increases by using PEF technology although the results differ notably depending on the olive cultivar and the conditions selected for the extraction, especially malaxation times and temperatures [15,16,17,18]. Significant increases in the content of bioactive components in the oils have also been reported [15,16]. However, some important discrepancies may be found among the results obtained by different authors, which in most cases used lab scale extraction systems [16,17,19]. In this sense, the aim of this work is to study the effect of PEF technology to confirm the benefits of this technology before making any recommendations on its use in oil mills. To this end, a series of extraction trials were conducted at a pilot and industrial scale. The effects that this innovative technology could have on the yield and quality of the oil was carefully analyzed, as well as the consequences on key aspects of the extraction process, such as malaxation time and temperature.

## 2. Materials and Methods

### 2.1. Plant Materials and Tests

Olive fruits from trees of cultivars ‘Manzanilla’ and ‘Hojiblanca’, grown in Hacienda Guzmán (Seville, Spain), were harvested at the turning ripening stage and immediately transported to IG-CSIC Pilot Plant in Seville. In 2020, harvest season pilot scale extraction trials were conducted with ‘Manzanilla’ and ‘Hojiblanca’ fruits (2000 kg and 1000 kg, respectively). In 2021, fruits from cultivar ‘Manzanilla’ (8000 kg) were used for the industrial trials conducted in November.

### 2.2. Oil Extraction Plants

The oil extraction plant used in 2020 for pilot scale trials was an Alfa Oliver olive oil extraction system model A0-500-TOP, with a decanter separator MSPX403-TGP-61 with a capacity for 250–400 kg olive fruits/h (Alfa Laval Iberia S.A, Madrid, Spain). In order to gain additional information on the effects of the treatment during the first extraction trials, olive pastes were sampled at different times along malaxation (0, 15 and 45 min) and centrifuged in an Abencor System (Abengoa, S.A., Seville, Spain) to analyze the changes induced by PEF in olive oils and pomaces.

The oil extraction plant used in 2021 for industrial extraction trials was a Pieralisi Integral Continuous System SPI-7 5E composed of different modules: a double grid crusher (1400 rpm, 38–32 Kw); a malaxer (Pieralisi Mod. 1250-2C) with two fully independent three-axis-malaxing units; and a two-phase decanter (Pieralisi Mod. SPI 7) with capacity for 4000 kg/h and a vertical centrifuge (Pieralisi SP-6000). Full information on the fruits used and the operation parameters of each extraction trial is shown in Table 1.

### 2.3. PEF Treatment

The PEF treatment was applied with a semiconductor-based positive Marx modula-tor [20]. For semi-pilot scale line (400 kg/h), model EPULSUS-PM1–10 (Energy Pulse Systems, Lisbon, Portugal), with a maximum pulse voltage of 10 kV, maximum pulse current of 200 A and 3 kW output average power, equipped with a DN25 colinear treatment chamber, with 2.5 cm between the electrodes was used. For the industrial scale line (4000 kg/h), model EPULSUS-PM2B-10 (EnergyPulse Systems, Lisbon, Portugal) with a maximum pulse voltage of 10 kV, maximum pulse current of 400 A and 6 kW out-put average power, equipped with a DN50 colinear treatment chamber, with 5 cm between the electrodes was used. On both scales, the applied electric field was 2 kV/cm in all trials with a 30 µs pulse width at a frequency of 90 Hz. The measured pulse current was approximately 50 A for an applied specific energy of 5 kJ/kg to olive. In all the extraction trials the PEF device was installed between the mill and the malaxer. Control oils were obtained with the PEF device installed but disconnected.

### 2.4. Oil Yield

The yield of the extraction process was calculated by using two different methods: (1) mass balance (kg of oil produced per kilo of olive fruit processed) and (2) analysis of the remaining oil content in the olive pomace generated in the extraction process by NMR using a minispec mq 100 Bruker apparatus (Billerica, MA, USA) [21]. These measurements were performed in triplicate.

### 2.5. Olive Oil Characterization

#### 2.5.1. Physicochemical and Sensory Analysis

The free acidity, peroxide value and spectrophotometric absorptions (K_232_, K_270_) of the oils were determined according to the official method for olive oil [22]. The organoleptic assessment was conducted by Instituto de la Grasa accredited panel (UNE-EN-ISO/IEC 17025) using the standard protocol of IOC [23].

#### 2.5.2. Oxidative Stability

The oxidative stability of the oils was evaluated by means of the Rancimat apparatus (Mod. 743, Metrohm Ltd., Herisau, Switzerland) at 120 °C using an air flow of 20 L/h. Stability was expressed as the oxidation induction time in hours.

#### 2.5.3. Analysis of Tocopherols

Tocopherols were determined by HPLC with fluorescence detection following IUPAC Standard Method 2.432 [24]. The oil samples were dissolved in n-heptane at a concentration of 50 mg mL^−1^ and analyzed in an Agilent 1260 Infinity HPLC chromatograph (Agilent Technologies, Santa Clara, CA, USA). The chromatograph was equipped with a quaternary pump VL (G1311C), a standard autosampler (G1329B), a thermostatted column compartment (TCC) (G1316A) and a fluorescence detector (G1321A). A silica HPLC column (LiChrospher^®^ Si 60, 250 mm × 4 mm i.d., 5 µm particle size) (Merck, Darmstadt, Germany) was used. The volume of the sample analyzed was 20 µL. The temperature of the TCC was set at 25 °C. The separation of tocopherols was performed using n-heptane:isopropanol (99:1, *v*/*v*) with a flow rate of 1 mL min^−1^. The excitation and emission wavelengths in the detector were 290 nm and 330 nm, respectively. Quantification was made by external calibration using tocopherol standards.

#### 2.5.4. Extraction and Analysis of Phenolic Compounds

VOO phenolic compounds were isolated by solid phase extraction (SPE) on a diol-bonded phase cartridge (Supelco, Bellefonte, PA, USA) based on the method by Mateos et al. [25] using *p*-hydroxyphenylacetic and *o*-coumaric acids as internal standards. Phenolic compounds were analyzed by HPLC on a Beckman Coulter liquid chromatography system equipped with a System Gold 168 detector, a solvent module 126 and a Waters column heater module following a previously described methodology [26]. A Superspher RP 18 column (4.6 mm i.d. × 250 mm, particle size 4 µm: Dr Maisch GmbH, Ammerbuch, Germany) at flow rate 1 mL min^−1^ and a temperature of 35 °C was used. Tentative identification of compounds was conducted with their UV-Vis spectra and later confirmed with HPLC/ESI-qTOF-HRMS on a liquid chromatograph Dionex Ultimate 3000 RS U-HPLC liquid chromatograph system (Thermo Fisher Scientific, Waltham, MA, USA) equipped with a similar column and elution program. Mass spectra were acquired in MS fullscan mode and data were processed using TargetAnalysis 1.2 software (Bruker Daltonics, Bremen, Germany).

#### 2.5.5. Extraction and Analysis of Volatile Compounds

VOO samples (0.5 g of oil in 10 mL vials) were placed in a vial heater at 40 °C and after a 10 min equilibration, volatile compounds were adsorbed onto a SPME fiber DVB/Carboxen/PDMS 50/30 μm (Supelco Co., Bellefonte, PA, USA) for 50 min. Desorption of volatile compounds was performed directly into the injector on a HP-6890 gas chromatography apparatus (Agilent Technologies, Santa Clara, CA, USA), equipped with a DB-Wax capillary column (60 m × 0.25 mm i.d., film thickness, 0.25 μm, J&W Scientific, Folsom, CA, USA). Three vials per sample were analyzed. Identification of volatile compounds was conducted with a 7820A/GC-5975/MSD system (Agilent Technologies, Santa Clara, CA, USA) equipped with the same column and using similar conditions. Full information on the operating conditions used for both systems 7820A/GC-5975/MSD and HP-6890, as well as the quantification method have been fully described in a previous paper [27]. The volatile compounds were individually quantitated and also clustered into different groups according to their origin in the lipoxygenase (LOX) pathway branch (C6 and C5 compounds) from linoleic (LA) and linolenic (LNA) acids, as well as terpenes and branched-chain (BC) volatile compounds from amino acid metabolism:

C6/LnA aldehydes: (*E*)-hex-2-enal, (*Z*)-hex-3-enal, (*Z*)-hex-2-enal, (*E*)-hex-3-enal.

C6/LnA alcohols: (*E*)-hex-2-enol, (*Z*)-hex-3-enol, (*E*)-hex-3-enol.

C6/LA aldehyde: hexanal.

C6/LA alcohol: hexan-1-ol.

C5/LNA carbonyls: pent-1-en-3-one, (*E*)-pent-2-enal, (*Z*)-pent-2-enal.

C5/LNA alcohols: pent-1-en-3-ol, (*E*)-pent-2-en-1-ol, (*Z*)-pent-2-en-1-ol.

PD: pentene dimers (seven isomers).

C5/LA carbonyls: pentan-3-one, pentanal.

C5/LA alcohol: pentan-1-ol.

LOX esters: hexyl acetate, (*Z*)-hex-3-en-1-yl acetate, (*E*)-hex-2-en-1-yl acetate.

Non-LOX esters: methyl acetate, ethyl acetate, methyl hexanoate, ethyl hexanoate.

Terpenes: limonene, β-ocimene.

BC aldehydes: 3-methyl-butanal, 2-methyl-butanal.

BC alcohol: 2-methyl-butan-1-ol.

#### 2.5.6. Statistical Analysis

Data were statistically evaluated using Statistica (Statsoft Inc., Tulsa, OK, USA), applying an analysis of variance (ANOVA) and comparing the means with the Student–Newman–Keuls/Duncan test. Significance was accepted at a level of 0.05.

## 3. Results

### 3.1. Effect of PEF Technology on Oil Yield

As previously mentioned in the introduction, increasing malaxation time and temperature up to a certain degree increases oil extraction yield but negatively affects oil quality. In this sense, new extraction technologies focus on reducing malaxation time and temperature without reducing the oil yield [12]. Thus, in the first pilot trials, the effect of applying a PEF treatment in combination with a reduced malaxation stage (45 min) was compared to the standard industrial protocol with longer malaxation times, 90 min. for ‘Manzanilla’ fruits at turning stage (Table 1). Under these conditions, the standard process (control) gave rise to an oil yield 1% higher than the PEF process, using any of the methods to evaluate the efficiency of the extraction, mass balance or quantification of the remaining oil content in the olive pomaces (Table 2). However, when a short malaxation time was fixed (30 min) to compare exclusively the effect of PEF technology on the oil extraction process, it was observed that in general a higher oil yield was obtained in the pilot extractions conducted with PEF technology on both cultivars ‘Manzanilla’ and ‘Hojiblanca’ (Table 2). The best results in terms of oil yield were obtained in the industrial trials conducted with ‘Manzanilla’ fruits, in which the oil content of the olive pomace obtained was reduced by about 23% (from 16.6 to 12.7%) by applying PEF technology compared to the control process. Previous studies had already highlighted the potential of the PEF to increase the oil yield. However, in many of these studies oil extractions were conducted with a lab scale. Thus, increases of up to 1.7% oil yield have been described from ‘Arbequina’ fruits in tests conducted with a lab scale [19]. Working on a similar scale (close to 1 kg of olive fruits extraction), oil yield increases of around 1% have also been described for the Greek olive cultivars ‘Anfisis’ and ‘Manaki’ although the application of PEF technology had no effect on the olive cultivar ‘Tsounati’ [16]. At semi-pilot and pilot scales, positive results have been obtained with some Italian cultivars such as ‘Carolea’, ‘Coratina’ and ‘Nocellara del Belice’ [18,28] though negative results were obtained with cultivar ‘Ottobratica’ in similar conditions. These results and those that we have found in terms of oil yield with the cultivars ‘Hojiblanca’ and ‘Manzanilla’ suggest that it seems reasonable that the use of PEF technology must be specifically adjusted to the different characteristics of the olive fruits that arrive at the olive oil extraction plant.

Processing temperature seems also to be a critical point according to differences observed in terms of oil yield in the extraction trials conducted with ‘Manzanilla’ fruits. Thus, the oil extraction yield is inversely proportional to the malaxation temperature used, showing higher yields when the extraction was conducted below 20 °C (Table 1 and Table 2). Similarly, Abenoza et al. [19] reported significant increases in oil yields in samples treated with PEF technology and extracted at 15 °C, but not when the experiments were conducted at 25 °C. More recently, Andreou et al. [17] also concluded that the best results in terms of oil yield and quality were obtained when low malaxation temperatures were used in the combined application of PEF technology and high pressure on a lab scale. In relation to malaxation temperature, it is important to point out that only minor, non-significant increases of the temperature were detected in PEF-treated olive pastes (Table 1).

### 3.2. Effect of PEF Technology on the Trade Standards for Olive Oil

None or the parameters used to determine commercial categories of olive oils were significantly affected by PEF treatment. All the values obtained for free acidity, peroxide value, K_232_ and K_270_ were compatible with the ‘Extra virgin’ olive oil (EVOO) classification (Table 3). In a similar way, the sensory evaluation of the oils by using a panel test did not detect the appearance of any defect or off-flavor in the oils obtained by PEF treatment. In each extraction trial, both oil samples, control and PEF-treated, obtained the same classification. Thus, all ‘Manzanilla’ oils were classified as EVOO and both ‘Hojiblanca’ oils were classified as ‘Virgin’ with a ‘winey’ defect of 1.4 and 2.20 in control and PEF samples, respectively (Table 4). However, the intensity of positive attributes (fruity, bitter and pungent) was generally higher in PEF oils than in control oils. Other parameters, such as fatty acids, sterols and waxes profiles were also analyzed, but no significant changes were detected either (data not shown). Regarding the oxidative stability measured with Rancimat apparatus, in general the oils obtained with PEF technology showed longer oxidation induction times than those of control oils in most extraction tests (Table 3). However, differences were not statistically significant. Similar increases (2–10%) were also detected by Andreou et al. [16] using a lab scale PEF assisted extraction system and by Veneziani et al. [28] in ‘Coratina’ oils obtained by applying PEF technology after the malaxation phase.

### 3.3. Effect of PEF Technology on the Phenolic and Tocopherol Composition of VOO

In the previous sections it has been shown that the application of PEF technology to the industrial production of VOO can increase the oil extraction yield without affecting the VOO trade standards. However, the main objective of our study was to evaluate the possible benefits of the application of PEF technology to increase the functional and organoleptic quality of VOO. In this sense, the profile of phenolic compounds and tocopherols in the oils was analyzed.

The phenolic composition of the oils obtained from cultivars ‘Manzanilla’ and ‘Hojiblanca’ was clearly different (Table 5). The total phenolic contents of the ‘Manzanilla’ oils were always quite high, above 500 mg kg^−1^, while the phenolic contents of the ‘Hojiblanca’ oils were extremely low, well below the limit of 250 mg kg^−1^ of hydroxytyrosol and derivatives established by the EFSA health claim [6]. As shown in Table 5, the application of PEF technology in the oil extraction of the ‘Manzanilla’ cultivar resulted in significantly (*p* ≤ 0.05) higher contents of phenolic compounds in both the pilot and industrial extraction tests. On the contrary, no positive effect of the PEF treatment was observed on the phenolic content in the oils of the cultivar ‘Hojiblanca’.

In addition to analyzing the final phenolic content of the oils, different trials were conducted to determine the effect of malaxation time, after the application of PEF technology, on the extraction/formation of key phenolic compounds. In this sense, different trials with cultivar ‘Manzanilla’ were conducted at different malaxation times: 0, 30 and 45 min. Data obtained on the phenolic composition of the oils suggests that the accumulation kinetics of phenolic compounds in the oil are clearly different in PEF-treated olive paste than in control extractions. Figure 1 shows the content of oleacein (3,4-DHPEA-EDA) (Figure 1A) and oleocanthal (p-HPEA -EDA) (Figure 1B), perhaps the two phenolic components with the highest impact on the functional [29] and flavor properties of VOO [30]. At the beginning of the malaxation stage (time 0) PEF oils have approximately twice the concentration of both phenolic compounds than the control oils. After 30 min of malaxation, although the contents of oleacein and oleocanthal remained significantly higher in PEF oils, the differences with control oils were reduced to 50%. Finally, after 45 min of malaxation, both PEF-treated and control oils exhibited very similar contents. Thus, the amount of these secoiridoid derivatives in VOO extracted with PEF was very high already from the beginning of the malaxation stage, while in control oils those contents progressively increased along malaxation. 

These results suggest that the PEF treatment allows a more efficient and rapid extraction, biosynthesis and/or absorption of the main secoiridoid phenolics in the oil. Thus, the use of PEF technology would decisively enhance the incorporation of phenolic compounds into the oil at short malaxation times. On the contrary, the use of PEF technology in extraction processes with long malaxation times probably does not provide significant benefits in terms of phenolic content. According to these results, a malaxation time of 30 min after the PEF treatment of the olive pastes seems to be the best option in terms of phenolic content.

It is important to highlight that the phenolic content of VOO is determined by the content of phenolic glycosides present in the olive fruit and the activity of the endogenous enzymes β-glucosidase and polyphenol oxidase, which regulate the content of secoiridoid derivatives in the oil [2,9]. Thus, if due to genetic or environmental factors the olive fruits have a low phenolic content or the activity levels of key enzymes within the phenolic metabolism are not adequate, no technological treatment will allow obtaining oils with an optimal phenolic content from those fruits. According to the few published data on the effect of PEF technology on the phenolic composition of VOO, there seems to be a strong varietal component. Thus, significant increases in the phenolic content of VOO have been reported by applying PEF technology in cultivars ‘Arroniz’, ‘Coratina’, ‘Carolea’ and ‘Ottobratica’ [15,28]. Very slight or no effect has been found for other olive cultivars such as ‘Nocellara del Belice’ [18]. In our study, the application of PEF technology not only affected the total phenolic content of VOO, but also the content of some individual phenolic components. Thus, statistically significant increases (*p* ≤ 0.05) were found in the contents of oleacein, oleocanthal, 3,4-DHPEA-EA and hydroxytyrosol acetate. The contents of some of these phenolic compounds in the ‘Manzanilla’ oils extracted with PEF technology were in the range 10–17% higher than in control oils. In this sense, Tamborrino et al. [18] had already described similar results with the ‘Nocellara del Belice’ cultivar, finding significantly higher amounts of oleacein and oleocanthal compounds in the oils obtained with PEF technology, although without observing significant increases in the total phenolic content.

In contrast to the results described on the phenolic profile of VOO, the analysis of tocopherols revealed that PEF technology does not significantly affect the content of these compounds (Table 6). The different effect of PEF technology on phenolic compounds and tocopherols could be explained by the fact that tocopherols are trapped by the oil at the milling stage since they are very lipophilic, while the main phenolic derivatives of VOO are synthesized from hydrophilic precursors in the fruit during the oil extraction process. In this sense, previous studies have reported that the content of tocopherols in VOO seems to suffer little variation by the conditions of the extraction [31], with genetic and agronomic factors being the main factors that determine their final concentration in the oil.

### 3.4. Effect of PEF Technology on the Volatile Composition of VOO

Although there are obvious differences between the volatile profiles of different olive cultivars—and even within the oils of the same olive cultivar but from different harvesting seasons—the data obtained reveal some modifications that seem to be associated with the application of PEF technology on olive pastes (Appendix A). Probably the most important modification is that all PEF-treated oils, except for ‘Manzanilla’ oils in the second trial, always have higher (*E*)-hex-2-enal contents than their corresponding controls (Appendix A). This data is quite relevant given that this compound is qualitatively and quantitatively the most important component for the aroma of VOO due to its high concentration in most olive oils and its relatively low odor threshold [32]. In most of the PEF oils, significantly higher contents of C6/LA aldehydes and some C5 compounds were also found. The aroma of VOO is made up of a complex mixture of volatile compounds, in which aldehydes and alcohols with 5 (C5) and 6 (C6) carbon atoms and their corresponding esters formed through the LOX pathway stand out [10]. The LOX pathway begins with the release of linoleic and linolenic acids that are part of the cell membranes, and it is triggered by the crushing of the olive tissue. Thus, it seems reasonable that the application of the PEF treatment immediately after the milling step could increase plant cell disruption and enhance the formation of volatiles through the LOX pathway. In previous studies in which PEF technology was applied after the malaxation stage, when the LOX pathway is no longer active, no significant changes in the volatile composition of the oils have been observed [28]. On the contrary, Tamborrino et al. [18] also detected changes in the volatile composition of VOO when PEF technology was applied immediately after milling, before the malaxation stage. As previously described for phenolic compounds, we have evaluated the effect of PEF technology on the volatile composition of oils at different malaxation times. Figure 1C shows the content of (*E*)-hex-2-enal in control and PEF oils (cultivar ‘Manzanilla’) as a function of the malaxation time. As in the case of phenolics, the differences between both oils were greater the shorter the malaxation time. Thus, the content of (*E*)-hex-2-enal in PEF oils reached its maximum at time 0, while in control oils the content of this key volatile component increased steadily and reached its maximum concentration after 45 min. These data suggest that PEF technology would allow a more efficient and faster biosynthesis of this compound and/or absorption of this compound by the recently released oil, as happened with the phenolic compounds oleacein and oleocanthal. These results are probably associated with a stronger induction of the LOX pathway due to the additional damage caused by PEF in cell membranes, which would allow a greater release of substrates, such as the unsaturated fatty acids LA and LNA and enzymes such as the hydroperoxyde lyase that are strongly associated to plant cell membranes.

## 4. Conclusions

The results obtained in pilot and industrial scale trials suggest that PEF technology could have positive effects on the oil yield and could improve the extraction of positive minor components such as phenolic and volatile compounds. The results seem to be conditioned by the olive cultivar and ripening stage of the fruits and also by the processing conditions. The best results were obtained by combining the PEF treatment (2 kV/cm) with short malaxation times and temperatures (below 20 °C). With these processing conditions, using ‘Manzanilla’ fruits harvested at the optimum ripening point, PEF technology can increase oil yield by up to 25%; it can also favor the enrichment of phenolic and volatile compounds in the oil. The PEF treatment neither affected the physicochemical parameters used to determine the commercial categories of olive oils, nor the tocopherol content. Similarly, the sensory evaluation of the PEF-extracted oils by using a ‘panel test’ did not detect any defects or off-flavors attributable to the treatment.

## Figures and Tables

**Figure 1 foods-11-02022-f001:**
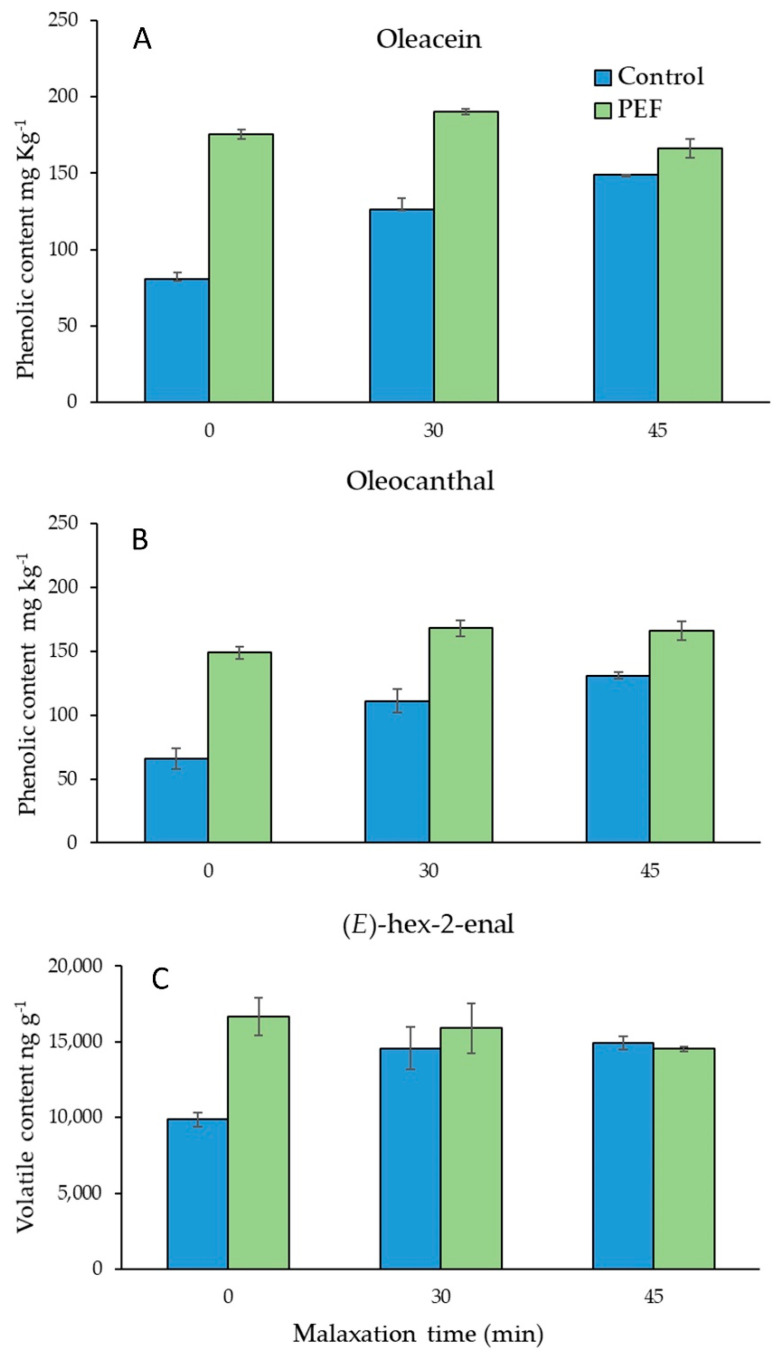
Content of oleacein, oleocanthal and (*E*)-hex-2-enal in control and PEF oils obtained from ‘Manzanilla’ fruits using different malaxation times: 0, 30 and 45 min.

**Table 1 foods-11-02022-t001:** Content of water and oil of olive fruits and operating parameters for the extraction trials conducted with fruits of cultivars ‘Manzanilla’ and ‘Hojiblanca’ extracted with a standard protocol (Control) and by applying pulsed electric field (PEF) technology.

	1st Pilot Tests		2nd Pilot Tests		3rd Pilot Tests		Industrial Tests	
	Control	PEF	Control	PEF	Control	PEF	Control	PEF
Olive fruits								
Olive cultivar	‘Manzanilla’		‘Manzanilla’		‘Hojiblanca’		‘Manzanilla’	
Water (%)	56.18	56.45	61.49	61.13	61.15	65.55	62.61	63.10
Oil (% dry weight-RMN)	43.61	43.29	44.4	42.51	45.51	46.34	44.79	45.55
Oil (% fresh weight-RMN)	19.11	18.85	17.09	16.52	15.67	15.96	16.05	15.89
Operation parameters								
Malaxation time (min)	90	45	30	30	30	30	30	30
Decanter mass flow (kg h^−1^)	384	384	310	310	318	384	2400 kg/h	2400
Decanter water flow (L h^−1^)	0	0	0	0	0	0	0	0
Temperature before PEF (°C)	28	28	22	22	21.2	18.3	16	16.4
Temperature after PEF (°C)	30	30	24	24	21.8	21.9	17	18.1
Oil flow vertical centrifuge (kg h^−1^)	53.8	49.7	32.8	35.5	29.9	33.3	432.0	420.0
Water flow in VC (L h^−1^)	20	20	20	20	20	20	100	101

**Table 2 foods-11-02022-t002:** Oil extraction yields and oil contents of pomaces obtained from ‘Manzanilla’ and ‘Hojiblanca’ olive fruits extracted with a standard protocol (Control) and by applying pulsed electric field (PEF) technology.

		Oil Yield	Olive Pomace
		(% *w*/*w* Fruit)	(Oil % Dry)
1st pilot test	Manzanilla Control	13.9	15.5
	Manzanilla PEF	12.9	16.8
2nd pilot test	Manzanilla Control	10.6	18.3
	Manzanilla PEF	11.4	18.5
3rd pilot test	Hojiblanca Control	9.6	21.2
	Hojiblanca PEF	10.7	20.2
Industrial test	Manzanilla Control	-	16.6
	Manzanilla PEF	-	12.7

**Table 3 foods-11-02022-t003:** Trade standards for olive oils from cultivars ‘Manzanilla’ and ‘Hojiblanca’ extracted with a standard protocol (Control) and by applying pulsed electric field (PEF) technology.

		Acidity (% Oleic Acid)	Peroxide Value (meq O_2_ kg^−1^)	K_232_	K_270_	Rancimat (h)
1st pilot tests	Manzanilla Control	0.20	5.71	1.65	0.12	13.45
	Manzanilla PEF	0.20	5.07	1.59	0.12	14.16
2nd pilot tests	Manzanilla Control	0.16	6.39	1.70	0.13	12.81
	Manzanilla PEF	0.16	5.33	1.66	0.13	13.71
3rd pilot tests	Hojiblanca Control	0.24	10.63	1.91	0.11	4.90
	Hojiblanca PEF	0.22	9.80	1.86	0.11	5.13
Industrial tests	Manzanilla Control	0.37	4.51	12.85	1.52	12.85
	Manzanilla PEF	0.37	4.53	11.62	1.52	11.62

**Table 4 foods-11-02022-t004:** Sensory evaluation of control and PEF oils obtained from ‘Manzanilla’ and ‘Hojiblanca’ cultivars.

		Fruity	Bitter	Pungent	Defects	Classification
1st pilot tests	Manzanilla Control	3.8	3.8	4.2	0.0	Extra virgin
	Manzanilla PEF	4.4	4.4	5.3	0.0	Extra virgin
2nd pilot tests	Manzanilla Control	4.4	3.3	4.8	0.0	Extra virgin
	Manzanilla PEF	4.6	4.5	5.1	0.0	Extra virgin
3rd pilot tests	Hojiblanca Control	3.6	2.6	3.4	1.4 (winey)	Virgin
	Hojiblanca PEF	2.8	3.1	3.8	2.2 (winey)	Virgin
Industrial tests	Manzanilla Control	4.3	4.85	4.7	0.00	Extra virgin
	Manzanilla PEF	4.45	4.10	5.1	0.00	Extra virgin

**Table 5 foods-11-02022-t005:** Phenolic composition of control and PEF oils obtained from ‘Manzanilla’ and ‘Hojiblanca’ cultivars.

	1st Pilot Tests		2nd Pilot Tests		3rd Pilot Tests		Industrial Tests	
Phenols (mg kg^−1^)	Control	PEF	Control	PEF	Control	PEF	Control	PEF
Hydroxytyrosol	1.3 ± 0.4	1.2 ± 0.2	0.0 ± 0.8	1.5 ± 0.3	0.6 ± 0.0	1.2 ± 0.8	2.0 ± 0.1	2.3 ± 0.2
Tyrosol	2.3 ± 0.1	2.1 ± 0.0	3.3 ± 0.2	3.1 ± 0.1	5.0 ± 0.4	5.0 ± 0.6	4.5 ± 0.1	4.7 ± 0.0
Vanillic acid	0.4 ± 0.1	0.3 ± 0.0	0.5 ± 0.1	0.4 ± 0.0	0.6 ± 0.1	0.7 ± 0.0	0.6 ± 0.0	0.6 ± 0.1
Vainillin	0.1 ± 0.0	0.2 ± 0.0	0.1 ± 0.0	0.1 ± 0.0	0.1 ± 0.1	0.1 ± 0.0	0.2 ± 0.0	0.2 ± 0.0
*p*-coumaric acid	1.3 ± 0.2	1.2 ± 0.3	0.9 ± 0.2	0.9 ± 0.1	0.7 ± 0.1	0.6 ± 0.3	0.8 ± 0.0	0.8 ± 0.0
Hty acetate	3.7 ± 0.2	6.2 ± 0.3 *	1.9 ± 0.0	2.0 ± 0.0	1.2 ± 0.2	6.1 ± 7.8	2.4 ± 0.2	2.4 ± 0.5
3.4-DHPEA-EDA	126.3 ± 4.2	147.4 ± 7.4 *	59.0 ± 3.8	69.5 ± 4.5 *	9.1 ± 0.9	8.2 ± 3.4	66.6 ± 0.08	61.8 ± 0.1
*p*-HPEA-EDA	147.1 ± 2.9	164.6 ± 6.5 *	74.6 ± 3.2	75.8 ± 5.8	26.1 ± 1.8	21.9 ± 8.4	31.4 ± 1.5	26.5 ± 0.9
Pinoresinol	7.7 ± 0.2	7.4 ± 0.3	6.7 ± 0.2	9.1 ± 0.4	3.0 ± 0.1	2.5 ± 0.6	0.3 ± 0.0	0.2 ± 0.0
Cinnamic acid	0.6 ± 0.5	0.5 ± 0.4	3.4 ± 0.1	2.6 ± 0.1	0.3 ± 0.1	0.9 ± 0.4	0.9 ± 0.0	0.9 ± 0.0
Acetoxypinores	10.7 ± 3.8	10.8 ± 2.8	25.4 ± 3.4	27.0 ± 2.9	15.7 ± 0.8	13.1 ± 2.9	23.6 ± 1.2	23.5 ± 2.0
3.4-DHPEA-EA	176.2 ± 3.9	172.5 ± 6.1	301.5 ± 13.0	337.0 ± 17.6 *	34.0 ± 2.6	30.0 ± 4.1	362.1 ± 9.1	395.2 ± 3.6 *
*p*-HPEA-EA	50.0 ± 0.8	49.5 ± 1.6	42.9 ± 1.1	41.5 ± 7.7	6.4 ± 0.3	6.3 ± 1.2	24.7 ± 1.6	22. 9 ± 0.9
Ferulic acid	0.3 ± 0.0	0.3 ± 0.0	0.3 ± 0.1	0.3 ± 0.1	0.2 ± 0.0	0.2 ± 0.0	0.5 ± 0.0	0.5 ± 0.00.0
Luteolin	2.0 ± 0.2	1.8 ± 0.0	3.5 ± 0.6	3.10 ± 0.7	10.3 ± 1.1	10.1 ± 1.5	2.2 ± 0.0	2.3 ± 0.10.1
Apigenin	0.7 ± 0.2	0.6 ± 0.1	1.4 ± 0.5	1.1 ± 0.1	3.4 ± 0.6	3.3 ± 0.7	0.9 ± 0.0	0.8 ± 0.1
Total phenolics	530.9 ± 7.3	566.7 ± 20.1 *	526.9 ± 22.3	575.1 ± 40.1 *	117.0 ± 3.2	110.2 ± 25.7	523.7 ± 8.5	545.7 ± 0.6 *
Total *o*-phenolics	309.6 ± 7.8	329.1 ± 14.0	367.5 ± 17.1	413.1 ± 23.1 *	55.4 ± 2.5	55.6 ± 14.6	435.4 ± 9.9	464.1 ± 3.3 *
Total secoiridoids	499.6 ± 9.7	534.1 ± 21.4	477.9 ± 18.9	523.8 ± 35.5 *	75.7 ± 4.5	66.4 ± 15.7	484.9 ± 9.9	506.5 ± 1.7 *

* Significantly higher (*p* ≤ 0.05) than that of control oil.

**Table 6 foods-11-02022-t006:** Tocopherols content (mg kg^−1^) in control and PEF oils obtained from ‘Manzanilla’ and ‘Hojiblanca’ cvs.

Tocopherols (mg kg^−1^)		α-toc	β-toc	γ-toc	Total
1st pilot tests	Manzanilla Control	152.50 ± 0.17	1.40 ± 0.00	9.10 ± 0.16	163.00 ± 2.60
	Manzanilla PEF	153.10 ± 0.02	1.60 ± 0.00	9.20 ± 0.01	164.00 ± 1.90
2nd pilot tests	Manzanilla Control	185.50 ± 0.28	1.80 ± 0.00	10.10 ± 0.80	199.90 ± 1.70
	Manzanilla PEF	187.50 ± 0.09	1.80 ± 0.00	10.00 ± 0.08	199.30 ± 0.40
3rd pilot tests	Hojiblanca Control	303.00 ± 0.21	4.10 ± 0.00	12.50 ± 0.19	319.50 ± 0.90
	Hojiblanca PEF	304.40 ± 0.12	4.10 ± 0.00	12.80 ± 0.02	321.30 ± 0.80
Industrial tests	Manzanilla Control	131.57 ± 0.05	1.00 ± 0.00	21.32 ± 0.32	154.92 ± 1.80
	Manzanilla PEF	133.79 ± 0.01	1.00 ± 0.00	22.14 ± 0.32	157.46 ± 1.14

Results are expressed as means ± SD (*n* = 3).

## Data Availability

Data is contained within the article.

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
