# Peer review of "Application of Pulsed Electric Fields to Pilot and Industrial Scale Virgin Olive Oil Extraction: Impact on Organoleptic and Functional Quality"

_foods, 2022, doi:10.3390/foods11142022_

Round 1
Reviewer 1 Report
- Line 59: replace "are" with "is"
- Table 1 shows the oil content data using a method that is described in the following paragraphs. Would it be better to place it after the description of the methods? Table 1 shows the oil content data using a method that is described in the following paragraphs.
- The units of measurement adopted must be homogeneous throughout the text (including tables and figures). For example kg/h e Kg/h should be replaced with kg h-1; meq O2/kg with meq O2 kg-1
- Throughout the text replace 'Noccelara del Belice' with 'Nocellara del Belice'.
- Line 276: replace 'Extra-virgin' with 'Extra virgin'
- Line 281: I am not aware that the intensities of the winey defect are similar
- Line 289: replace 'Veneciani' with 'Veneziani'
- Table 3: replace '(% oleic)' with '(% oleic acid)'; 'K232 and K270' with 'K232 and K270'; significant digits for acidity must always be two decimal places: 0.20 not 0.2
- Table 4: replace 'Extra-virgin' with 'Extra virgin'; all medians must be reported to one decimal place: 0.0 not 0, 1.4 not 1.40
- Throughout the text replace 'ppm' with 'mg kg-1'
Author Response
Answers to reviewer 1
Line 59: replace "are" with "is"
Done
- Table 1 shows the oil content data using a method that is described in the following paragraphs. Would it be better to place it after the description of the methods? Table 1 shows the oil content data using a method that is described in the following paragraphs.
Following the reviewer's suggestion table 1 has been moved and placed after section 2.4
- The units of measurement adopted must be homogeneous throughout the text (including tables and figures). For example kg/h e Kg/h should be replaced with kg h-1; meq O2/kg with meq O2 kg-1
Units format has been revised in Table 1 and throughout the text
- Throughout the text replace 'Noccelara del Belice' with 'Nocellara del Belice'.
Done.
- Line 276: replace Extra virgin' with 'Extra virgin'
Done.
- Line 281: I am not aware that the intensities of the winey defect are similar
Winey intensities were 1.4 and 2.2 for control and PEF ‘Hojiblanca’ oils. They are different but in both cases it is a low intensity defect, within the first quartile of the scale. In any case, in accordance with the reviewer's suggestion, the text has been modified, eliminating the word "similar" and indicating the specific values of the defect in both oils
- Line 289: replace 'Veneziani' with 'Veneziani'
Done.
- Table 3: replace '(% oleic)' with '(% oleic acid)'; 'K232 and K270' with 'K232 and K270'; significant digits for acidity must always be two decimal places: 0.20 not 0.2
- Table 4: replace Extra virgin' with 'Extra virgin'; all medians must be reported to one decimal place: 0.0 not 0, 1.4 not 1.40
- Throughout the text replace 'mg kg-1' with 'mg kg-1'
Done
Reviewer 2 Report
Application of pulsed electric fields to pilot and industrial scale virgin olive oil extraction: impact on organoleptic and functional quality
I have published a lot of work on the PEF but this is a unique work, and it is also difficult for me to review this paper. Great work and I have these comments which will help you to improve the quality of the paper.
Comments to authors:
Line no 16: Why the PEF is selected for this extraction? Reason
Line no 22: Does PEF give only positive results or any adverse effect on oil composition?
Line no 27: Abstract is not showing the whole theme of research. Give some more elaborations.
Line no 37: Which nutritional compounds? Give some examples
Line no 41: Which health promoting property is mainly relat6ed with VOO?
Line no 76. Please provide the reference for this statement. Please see these papers: Innovative Food Science & Emerging Technologies, 60, 2020, 102309, Molecules, 26(16), 4893, Food Chemistry, 167, 2015, 497-502 & Trends in Food Science and Technology, 111(5), 43-54.
Line no 86. Please check this paper, I hope you will get some latest information from this paper regarding extraction of oil through these nonthermal techniques: Food Reviews International, 38(6), 1166-1196.
Line no 81: concern with these papers for more data
Line no 84: Does PEF effects any nutritional attribute adversely?
Line no 91: Increase in oil yield also compromise on quality?
Line no 99: introduction is very long trying to write it in a concise manner.
Line no 122. Statistical alphabets are missing in Table 1.
Line no 128: Tables data is very commendable.
Line no 368: Graphical representation is appreciable too.
Line no 369. (E)-hex-2-enal or (E)-hexenal or 2 hexanal??
Line no 378: which genomic and agronomic factor contributes mainly?
Line no 422: Results and discussion section is very comprehensive and well written but according to me at the end there should be a section of results in which the whole research theme should be discussed in a single paragraph. So it would be easy for others the get the whole idea of research.
Line no 437. Please correct the title in the reference
Other comments:
Introduction is very long, it should be reduced.
Statistical analysis not needed before results.
Statistical alphabets are missing in Table 1 and 5.
Author Response
Answers to reviewer 2
Application of pulsed electric fields to pilot and industrial scale virgin olive oil extraction: impact on organoleptic and functional quality. I have published a lot of work on the PEF but this is a unique work, and it is also difficult for me to review this paper. Great work and I have these comments which will help you to improve the quality of the paper.
Comments to authors:
Line no 16: Why the PEF is selected for this extraction? Reason
PEF technology was selected among other options (ultrasound, cavitation...etc) based on scientific evidences that highlights that PEF only causes minimal increases in temperature. In this sense, it is generally accepted that the increase in olive paste temperature has a detrimental effect on the quality of the virgin olive oil (VOO). Given that the main objective of our study was to evaluate the possible benefits of the application of new extraction technologies to improve the quality of VOO, PEF was a promising option.
Line no 22: Does PEF give only positive results or any adverse effect on oil composition?
We have analyzed the most important components of the oil (fatty acids, phenolic compounds, tocopherols, volatile compounds,…) and we have not found any significant negative effect of PEF treatment on VOO composition.
Line no 27: Abstract is not showing the whole theme of research. Give some more elaborations.
More details on the research have been included in the revised version of the manuscript
Line no 37: Which nutritional compounds? Give some examples
The most important nutritionally valuable compounds are cited in the next line:
Tocopherols (VitE), Phenolic compounds, pigments (chlorophylls and carotenoids)
After the reviewer suggestion the text has been rewritten for clarification
Line no 41: Which health promoting property is mainly relat6ed with VOO?
The main health promoting property is described in the next line
“in particular with regard to its protective effects against the states of chronic inflammation and derived diseases [2]”
Reference [2] by Bernardini and Visioli exhaustively describes the connection between olive oil and health issues.
Line no 76. Please provide the reference for this statement. Please see these papers: Innovative Food Science & Emerging Technologies, 60, 2020, 102309, Molecules, 26(16), 4893, Food Chemistry, 167, 2015, 497-502 & Trends in Food Science and Technology, 111(5), 43-54.
A new reference has been included to support the statement
Zia, S.; Khan, Shabbir, M.A.; Maan, A.A.; Khan, M.; Nadeem, M.; Khalil, A.A.; Din, A., Aadil, R.A. An inclusive overview of advanced thermal and nonthermal extraction techniques for bioactive compounds in food and food-related matrices, Food Rev. Int, 2022 38, 1166-1196, DOI: 10.1080/87559129.2020.1772283
Line no 86. Please check this paper, I hope you will get some latest information from this paper regarding extraction of oil through these nonthermal techniques: Food Reviews International, 38(6), 1166-1196.
The original reference Kumari et al., 2018 has been substituted by the more recent publication suggested by the reviewer. Zia et al., 2022
Line no 81: concern with these papers for more data
Line no 84: Does PEF effects any nutritional attribute adversely?
As we have previously answered to comment on line 22, we have analyzed the most important components of the oil (fatty acids, phenolic compounds, tocopherols, etc…) and we have not found any significant negative effect of PEF on the oil composition or quality.
Line no 91: Increase in oil yield also compromise on quality?
Any technological innovation within the virgin olive oil extraction process must provide good oil yield without negatively affecting VOO quality. Both objectives are in our opinion equally important. Increasing oil yield at the expense of oil quality is easy to achieve, i.e. increasing the temperatures and specially the malaxation times in the traditional VOO industrial extraction process.
Line no 99: introduction is very long trying to write it in a concise manner.
The introduction has been revised
Line no 122. Statistical alphabets are missing in Table 1.
Table 1 is simply descriptive of the samples and technological parameters used in the study. Contains no data to compare between them. In no case were significant differences found in the batches of olives used for the control and PEF extraction.
Line no 128: Tables data is very commendable.
Line no 368: Graphical representation is appreciable too.
Line no 369. (E)-hex-2-enal or (E)-hexenal or 2 hexanal??
The correct name is (E)-hex-2-enal according to IUPAC, the full text has been revised accordingly.
Line no 378: which genomic and agronomic factor contributes mainly?
Findings indicated that tocopherol content and composition were highly dependent on cultivar and, to a lesser extent, on the year’s climate and the fruit ripening stage
Line no 422: Results and discussion section is very comprehensive and well written but according to me at the end there should be a section of results in which the whole research theme should be discussed in a single paragraph. So it would be easy for others the get the whole idea of research.
A paragraph with the conclusions has been included in the revised version of the manuscript
Line no 437. Please correct the title in the reference
Done
Other comments:
Introduction is very long, it should be reduced.
Statistical analysis not needed before results.
A brief description of the Statistical analysis methodology used has been included
Statistical alphabets are missing in Table 1 and 5.
The question on Table 1 was already answered
In table 5, The statistically significant differences between control and PEF samples are marked with “*”